# Larval surveys reveal breeding site preferences of malaria vector *Anopheles* spp. in Zanzibar City

**Kaeden K. Hill**[1]*, **Dickson Kobe**[2], **Narriman S. Jiddawi**[3,4], **Jonathan R. Walz**[4], **Katharina Kreppel**[5]

**1** Trinity College of Arts and Sciences, Duke University, Durham, North Carolina, United States of America, **2** Ifakara Health Institute, Bagamoyo, Tanzania, **3** Institute of Marine Sciences, University of Dar es Salaam, Zanzibar, Tanzania, **4** SIT-Graduate Institute, Brattleboro, Vermont, United States of America, **5** Institute of Tropical Medicine, Antwerp, Belgium

* Kaeden.hill@duke.edu

## Abstract

In Zanzibar City - the capital of the Zanzibar archipelago in Tanzania - the incidence of malaria has decreased over the past few decades due to standardized treatment protocols and public health interventions targeting adult mosquitoes. However, the incidence remains between 1–2%, and case numbers have increased over the past few years because of a continued influx of *Plasmodium* spp. from other malaria-endemic areas (including mainland Tanzania). Larviciding is a powerful tool to target mosquito populations and reduce the incidence of malaria. However, larvicidal strategies rely on knowledge of the breeding patterns of malaria vector mosquitoes. In Zanzibar City, no larval surveys have been done in the last few years. Our aim was to characterize *Anopheles* spp. breeding sites in Zanzibar City during the rainy season. We first conducted systematic larval surveys across 16 semi-permanent/permanent water bodies and 30 temporary water bodies. Then, we used principal component analysis and logistic regression to model the effects of physical/chemical parameters and rainfall on *Anopheles* presence. We found that *Anopheles* spp. prefer concrete, semi-permanent breeding sites with high levels of dissolved oxygen but are also found in natural sites after heavy rains. Our logistic regression model successfully predicted the presence of *Anopheles* larvae, achieving a positive predictive power of 65.7% and a negative predictive power of 88.8%. The data from our study suggest that *Anopheles* spp. have not yet adapted to more polluted breeding sites in Zanzibar City (as they have in some mainland locations). These results can inform targeted larvicidal strategies in Zanzibar City.

## Introduction

Malaria is a vector-borne disease caused by parasites of the genus *Plasmodium* that results in over 500,000 deaths annually [1]. Malaria is transmitted by female *Anopheles* spp. mosquitoes. Throughout Tanzania, malaria is one of the most wide-spread

**Data availability statement:** All relevant data are within the manuscript and its Supporting Information files.

**Funding:** The author(s) received no specific funding for this work.

**Competing interests:** The authors have declared that no competing interests exist.

mosquito-borne diseases, with at least 2.5 million cases reported in 2022 [1–3]. Tanzania contributes to 4% of the total malaria deaths worldwide [1].

Amid changing climate conditions and an increase in tourism, Zanzibar City - the largest city of the Zanzibar archipelago that includes Stone Town - has experienced a recent increase in malaria cases. In 2009, the Zanzibar Ministry of Health initiated a set of policies to eliminate malaria on the archipelago, including increased indoor spraying, a systematic distribution of insecticidal nets, and a surveillance and case-detection system [4,5]. Zanzibar has therefore been able to reduce malaria transmission between 2005 and 2015 [6]. However, the yearly malaria incidence rate of 2.7 cases per 1,000 people reported in 2017 had increased to 3.6 by 2021 [5,7]. The increase in cases is thought to be due to an expansion of tourism and a greater influx of people infected with *Plasmodium* from mainland Tanzania, inadvertently providing blood meals for the local mosquitoes [8,9]. Additional vector control interventions are needed to reduce the resident mosquito population and help to eliminate malaria. Understanding the local ecology of the *Anopheles* mosquitoes is crucial to inform such control strategies. Due to their low mobility and fully-aquatic nature, mosquito larvae that develop in freshwater bodies are effective targets for reducing the abundance of adult vector mosquitoes and overall malaria transmission [10]. However, the breeding habits of *Anopheles* mosquitoes in Zanzibar City have not been characterized.

While urbanization was originally thought to decrease the incidence of malaria, unplanned, rapid urbanization in sub-Saharan Africa has been accompanied by sustained malaria transmission [11–14]. Zanzibar City has expanded 3.8% geographically per year since 2004, leading to unplanned urbanization with poor integration of water drainage into the city infrastructure, potentially increasing mosquito breeding sites [15]. Therefore, urbanization has likely contributed to enhanced malaria transmission in recent years [16]. With a population of 800,010 people in 2023, and a population growth rate of 4.48%, this trend is likely to continue [17].

Every year, particularly during the long rains between March and May, the pooling of stagnant freshwater in the urban environment creates breeding sites, thus contributing to greater numbers of disease vectors. Despite some existing drainage of the streets of Zanzibar City, excess water runoff often leads to the expansion or flooding of natural water pools (see S1 Fig). Additionally, Zanzibar City, as a tourist destination, features aesthetic ponds and nonfunctional fountains that often contain stagnant water, creating breeding sites for mosquitoes (particularly within historic Stone Town). Over recent years, climate change has resulted in shorter but heavier rains during the March-May rainy season, likely increasing the number of potential urban mosquito breeding sites in Zanzibar City [18,19]. For larvicidal strategies to be most effective against *Anopheles* mosquitoes, knowledge of the defining characteristics of *Anopheles* breeding sites is important. However, *Anopheles* breeding habits vary by geographical location and season, making it difficult to rely solely on studies conducted in other areas or at other times [20–22].

Currently, larvicidal strategies are rarely used in Africa [23,24]. However, larvicidal strategies may complement existing vector control measures in Zanzibar City as indoor

residual spraying (IRS) and bednets lose effectiveness. This decrease in effectiveness is due to a shift in the malaria vector population from mainly *Anopheles gambiae* to the more adaptable *Anopheles arabiensis* [25]. The latter commonly bites humans outdoors, thereby avoiding bed nets and IRS [26]. Additionally, *Anopheles arabiensis* is becoming increasingly resistant to pyrethroid insecticides [27,28]. Notably, an *Aedes aegypti*-focused larval survey was conducted in Zanzibar City in 2018, which did not report any *Anopheles* larvae [29]. However, the study focused on artificial, temporary breeding site types known to harbor *Aedes* larvae [30]. To our knowledge, no preexisting data regarding *Anopheles* breeding sites in Zanzibar City are publicly available. We set out to address the lack of data by conducting larval surveys and breeding site characterizations for *Anopheles* larvae in Zanzibar City during the long rainy season. We assessed whether physicochemical parameters, hydro-period, and predator presence could be used to predict whether a water body contained *Anopheles* larvae. We found consistently high *Anopheles* larvae abundance in concrete, semi-permanent structures with high dissolved oxygen levels. Additionally, we found high *Anopheles* larvae abundance in natural semi-permanent areas with high dissolved oxygen levels after heavier rains. Our findings suggest that *Anopheles* mosquito breeding sites can be identified using specific characteristics in Zanzibar City, allowing for targeted larviciding as a strategy to reduce malaria transmission.

## Methods

### Study site

Zanzibar City (locally known as Stone Town) is a popular tourist destination located on the Western coast of Unguja, the largest island of the Zanzibar archipelago (1,666 km$^2$) off the coast of East Africa. The city is located at approximately 6˚10' S 39˚12' E at sea-level. The climate is tropical, with an average annual rainfall of approximately 1,521mm [31]. Zanzibar City (and the entire Zanzibar archipelago) experiences two rainy seasons: short rains in October-December and long rains from March-May. As of 2023, Zanzibar City has a population of 800,010 people, which represents approximately 44% of the total population of the Zanzibar archipelago (1.8076 million people) [32].

### Sampling locations

Between April 11, 2024, and May 1, 2024, 16 permanent or semi-permanent sites and 30 temporary sites were sampled for *Anopheles* mosquito larvae (Fig 1A). Sites were initially found through reconnaissance or asking community members during the first week of the study. Because only publicly available water bodies were surveyed (and few such water bodies

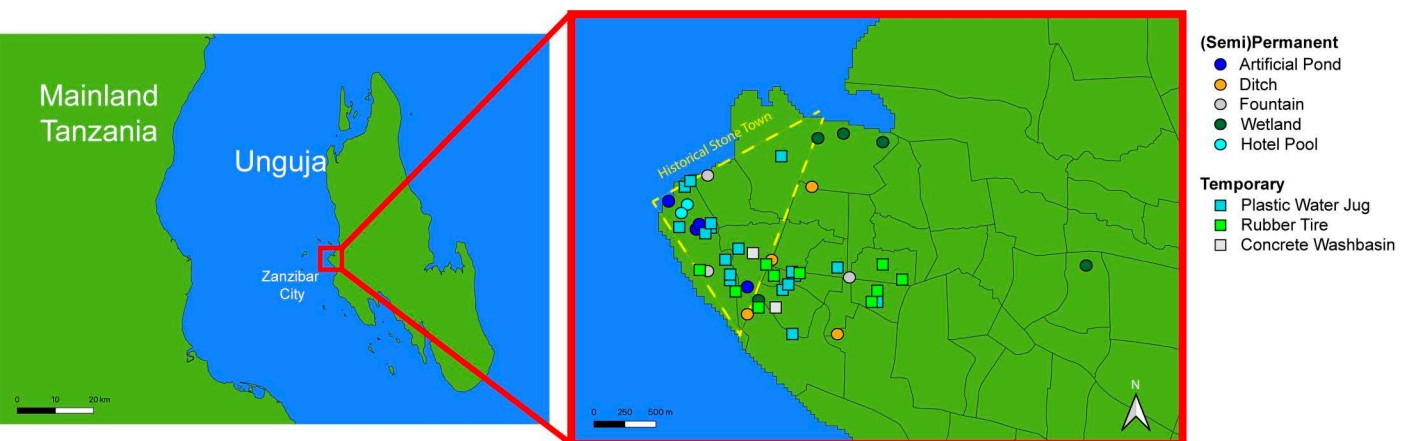

**Fig 1. Map of Zanzibar City, Tanzania, showing sites sampled for mosquito larvae.** Map generated using a shapefile provided by GADM under a CC BY license. The shapefile can be found at https://gadm.org/download_country.html.

were found in the city's Eastern sprawl), most study sites were near the more developed Stone Town and the densely populated neighborhoods nearby. Semi-permanent sites were defined as permanent structures that are occasionally drained (but rarely during the rainy season), while permanent sites were defined as structures that contain water continuously throughout the year. Semi-permanent and permanent sites sampled included unfinished or nonfunctional fountains (n = 3), artificial ponds (n = 4), roadside ditches (n = 3), and wetlands (n = 5) (S1 Fig). Temporary sites included tires (n = 10), plastic water containers (n = 18), and small concrete wash basins (n = 2). Each permanent or semi-permanent site larger than 4m in perimeter was divided into semi-randomized 0.5m x 0.5m subsites. A random number was generated that indicated the 0.5m x 0.5m site to sample. Before sampling, however, the 0.5m x 0.5m subsite was inspected to ensure that it could physically house mosquito larvae (e.g., adequate, stagnant water present), and that it was accessible with a larval dipper. If a randomly generated subsite could not be sampled, the nearest possible subsite was sampled instead. Once the subsite was identified, a 0.5m x 0.5m quadrat was placed into the water. At each site, data was collected at five subsites unless the site was smaller than five 0.5m x 0.5m quadrats (referred to herein as "subsite"). Because mosquito larvae distribution is not evenly distributed across larger habitats [33], each subsite was treated independently during analysis. Notably, all *Anopheles* species on Unguja are capable of transmitting *P. falciparum* [9,34], so mosquito larvae were identified to genus level. Mosquito genera on Unguja also include *Culex* and *Aedes* [35].

### Collection and identification of mosquito larvae

Mosquito larvae were sampled using a 500mL or 80mL dipper (depending on the size of the water body) and a white inspection tray. Five dips were conducted per 0.5m x 0.5m subsite, and the percentage of dips containing larvae of each mosquito genus was calculated. Dipping was performed by the same individual at all sites and visits. Other data collected for each dip included the number of mosquito larvae of each genus found per dip, and the number and name of other macroinvertebrates. Predators of mosquito larvae at the dipping site were also recorded.

Mosquito larvae were identified to the genus-level using a hand lens and identification keys [36,37]. Pupae were not recorded. If a particular larva could not be identified using a hand lens, the specimen was transferred to the laboratory for examination using an XSZ-107BN Series light microscope.

### Measurements of physical parameters

All physical parameters were measured within each 0.5m x 0.5m subsite before larvae were sampled. Coordinates were taken using the iPhone 11 compass. Outdoor and water temperature were both measured using a standard glass laboratory thermometer. Percentage vegetation cover was estimated by eye and included all living plants emerging at the surface of the subsite (e.g., patches of grass emerging from the water in a shallower subsite). Depth was measured within the subsite using either a metal measuring tape or a measuring tape with a weighted end. Other binary (yes or no) physical parameters recorded were: trash within 3m of the subsite, large dump (> 10 pieces of trash) within 3m of the subsite, semi-permanence of dipping site, and whether the dipping site was constructed out of concrete, plastic/rubber, or natural. Additionally, the perimeter was estimated and placed into one of five categories (1= < 1m, 2 = 1-5m, 3 = 5-10m, 4 = 10-20m, 5= > 20m). All daily precipitation data was obtained from the Visual Crossing website (accessed April 27, 2024), which compiles data collected at the HTZA weather station (-6.22, 39.22) and the Abeid Amani Karume International Airport weather station (-6.22, 39.23) approximately 5km from Zanzibar City [38].

### Measurements of chemical parameters

Dissolved oxygen concentration, dissolved oxygen saturation, pH, relative turbidity, and relative chlorophyll content were measured in each subsite on at least one of the visits for each site. Dissolved oxygen was measured on-site using a Ecosense DO200A within each 0.5m x 0.5m subsite before larvae were sampled. The instrument was calibrated before the first

use using the manufacturer's protocol. Both dissolved oxygen in parts per million (ppm) and percentage saturation were recorded. For the remaining chemical parameters, a water sample was collected in a cup (after first rinsing the cup in water from that subsite) and transported to the laboratory for immediate analysis. pH was measured using a Hanna HI991001 pH probe. In between measurements, the probe was briefly rinsed in deionized (DI) water and wiped off. The pH meter was calibrated before the first use using the manufacturer's protocol. Salinity was measured using a MarineDepot refractometer that was calibrated daily using DI water. Values for relative turbidity and chlorophyll content were obtained by taking photos of the water inside a clear cup using an iPhone 11 camera (with flash) and analyzing the color distribution using the color histogram function on Fiji ImageJ2 (2.14.0) [39]. The type of cup and the white background were kept constant across all photos. The mean color intensity for the red, green, and blue color channels in a selected area inside the cup were recorded and normalized to the mean values for a sample of DI water. The relative turbidity was calculated by adding the normalized mean intensities for the red, green, and blue channels together, and subtracting that value from three (since a total normalized color value of three indicates that the red, green, and blue channel intensities were identical to those of DI water and is therefore the maximum whiteness/clarity for water). Relative chlorophyll content was estimated using the ratio of normalized mean green intensity to normalized mean blue intensity, as this value captures both brighter greens (DI-like normalized green intensity, lower normalized blue intensity), and darker greens (lower normalized blue and green intensity).

## Statistical analyses

All statistical analyses were done using GraphPad Prism 10 (10.1.1). The Anderson-Darling test was used to test for normality, and either Mann-Whitney tests or Kruskal-Wallis tests with uncorrected Dunn stepdown tests were used to compare groups of unpaired non-parametric data. As each pairwise comparison stood alone, no corrections were used. All non-parametric, paired data was analyzed using a paired Wilcoxon test. Parametric, unpaired data was analyzed using an unpaired, two-tailed Student's T-test with Welch's correction. All categorical data was analyzed using either a Fisher's Exact test (for contingency tables with any frequencies below 10) or a Chi-Square test for independence. Principal component analysis was conducted to determine which parameters to further investigate using data standardized to a mean of zero and standard deviation of one. Principal components were selected using parallel analysis. While principal component regression (PCAR) was run on all physical predictors included in the study, a multiple logistic regression model was built using forward selection of parameters that were significant in the PCAR and chemical/interaction parameters for which there was evidence of a relationship. The interaction terms included in the multiple logistic regression model test whether concentration dissolved oxygen affects the presence of *Anopheles* differently if the subsite was semi-permanent, made from concrete, or if rainfall levels were higher, and whether rain within 48hrs affects the presence of *Anopheles* differently in concrete or in natural subsites. A significance level of 0.05 was used for all statistical tests. All maps were generated using Q-GIS (v3.24.2-Tisler) and shapefiles were from GADM (v4.1). Copyright (CC BY 4.0) license information can be found at https://gadm.org/license.html.

## Statement of ethics

Ethical approval for this study was granted by the World Learning Inc. Institutional Review Board, and a student research permit was granted by the Revolutionary Government of Zanzibar through the Second Vice President's Office. Although most land accessed was publicly owned, permission to sample was obtained from the relevant owners whenever a water body crossed into private property. No protected species were sampled.

## Inclusivity of global research

Additional information regarding the ethical, cultural, and scientific considerations specific to inclusivity in global research is included in the Supporting Information (S1 File).

## Results

### Characterization of physical properties of *Anopheles* breeding sites

To test whether *Anopheles* prefer permanent/semi-permanent sites or temporary sites in Zanzibar City during the rainy season, we compared the number of sites with *Anopheles* larvae between permanent/semi-permanent and temporary sites. We found that 56.25% of the permanent or semi-permanent sites contained *Anopheles* larvae at least once over the study duration, while only 10% of the temporary sites contained *Anopheles* larvae (Fisher's Exact Test, $P=0.0013$, OR = 11.57, 95% CI = 2.359–44.85) (S1 Table). The three temporary sites with *Anopheles* larvae included two plastic water containers and one rubber container. The permanent or semi-permanent *Anopheles* breeding sites were in both Stone Town and the greater Zanzibar City area, while the temporary *Anopheles* breeding sites were only found in the Zanzibar City area outside of Stone Town (Fig 2A). Pools were not included in the statistical analysis because they are subject to potentially confounding anthropogenic variables. However, three pool drains across two hotel pools in Stone Town were sampled for mosquito larvae, and no larvae were found. Additionally, six large rain tanks for drinking water were visually inspected for larvae (but not sampled due to ethical issues surrounding conducting larval dips in drinking water). Among the three tanks inspected that were left open, two had *Culex* larvae present. The three closed tanks inspected did not have any larvae.

*Anopheles* presence was associated with concrete, artificial, and permanent/semi-permerent subsites, in addition to water temperature and outdoor temperature (Fig 2B). *Culex presence* was associated with natural subsites and the presence of visible trash, while *Aedes* presence was associated with subsites made of plastic or rubber (Fig 2B). There was also statistical evidence for the permanent/semi-permanent artificial subsites harboring more *Anopheles* larvae than permanent/semi-permanent natural subsites (Two-tailed Mann-Whitney U test, $P=0.0022$, $U_{(93, 54)} = 1953$) (Fig 2C). Significantly more *Anopheles* larvae were caught at semi-permanent subsites than permanent subsites (Two-tailed Mann-Whitney U test, $P<0.0001$, $U_{(84, 63)} = 1701$) (Fig 2D). Cohabitation with other mosquito larvae genera was uncommon, and we found no statistical evidence of fewer *Anopheles larvae* in habitats shared with *Culex* or *Aedes* larvae (Fig 2E-F). Last, principal component analysis regression (PCAR) showed parameters with significant positive predictive capabilities, including permanence/semi-permanence, artificial, and concrete. Parameters with a significant negative predictive capability included *Culex* cohabitation, *Aedes* cohabitation, vegetation cover, the presence of a large dump within 3m, plastic/rubber, and natural. While the intercept was nonsignificant, the overall PCA regression was significant by analysis of variance ($F_{(3, 173)} = 3.288$, $P=0.0221$) (S2 Table).

### Characterization of chemical parameters of *Anopheles* breeding sites

Chemical parameters were associated with the presence of mosquito larvae of each genus. The percentage of dips per subsite with *Anopheles* was associated with dissolved oxygen concentration, dissolved oxygen saturation, and pH. The percentage of dips per subsite with *Culex* was associated with salinity, and the percentage of dips per subsite with *Aedes* was associated with relative turbidity and estimated chlorophyll content (Fig 3A). Although the distribution of the number of *Anopheles*, *Culex*, or *Aedes* larvae caught plotted against dissolved oxygen levels was not normally distributed (Anderson-Darling Test, $P<0.0001$), Gaussian curves were used to visualize which dissolved oxygen levels were associated with a higher larval abundance for each genus in permanent/semipermanent subsites. A least squares regression comparison of the Gaussian fits of larvae abundance plotted against dissolved oxygen saturation/concentration suggested that distributions differ between genera (Extra sum of squares F test, $F_{(6, 293)} = 6.897$, $P<0.0001$). The distribution of *Anopheles* larvae abundance was shifted towards higher dissolved oxygen levels than the distribution of *Culex* larvae abundance, while *Aedes* larvae were found in low abundance across a wide range of dissolved oxygen levels (Fig 3B-C). While a Gaussian curve could not be fit for larvae abundance plotted against pH, the distribution of *Anopheles* larvae abundance was shifted towards higher pH levels than *Culex* (Fig 3C). *Aedes* larvae were found in low abundance across a wide range of pH levels. Turbidity appeared to have little effect on the distribution of larvae abundance from different genera (Fig 3E).

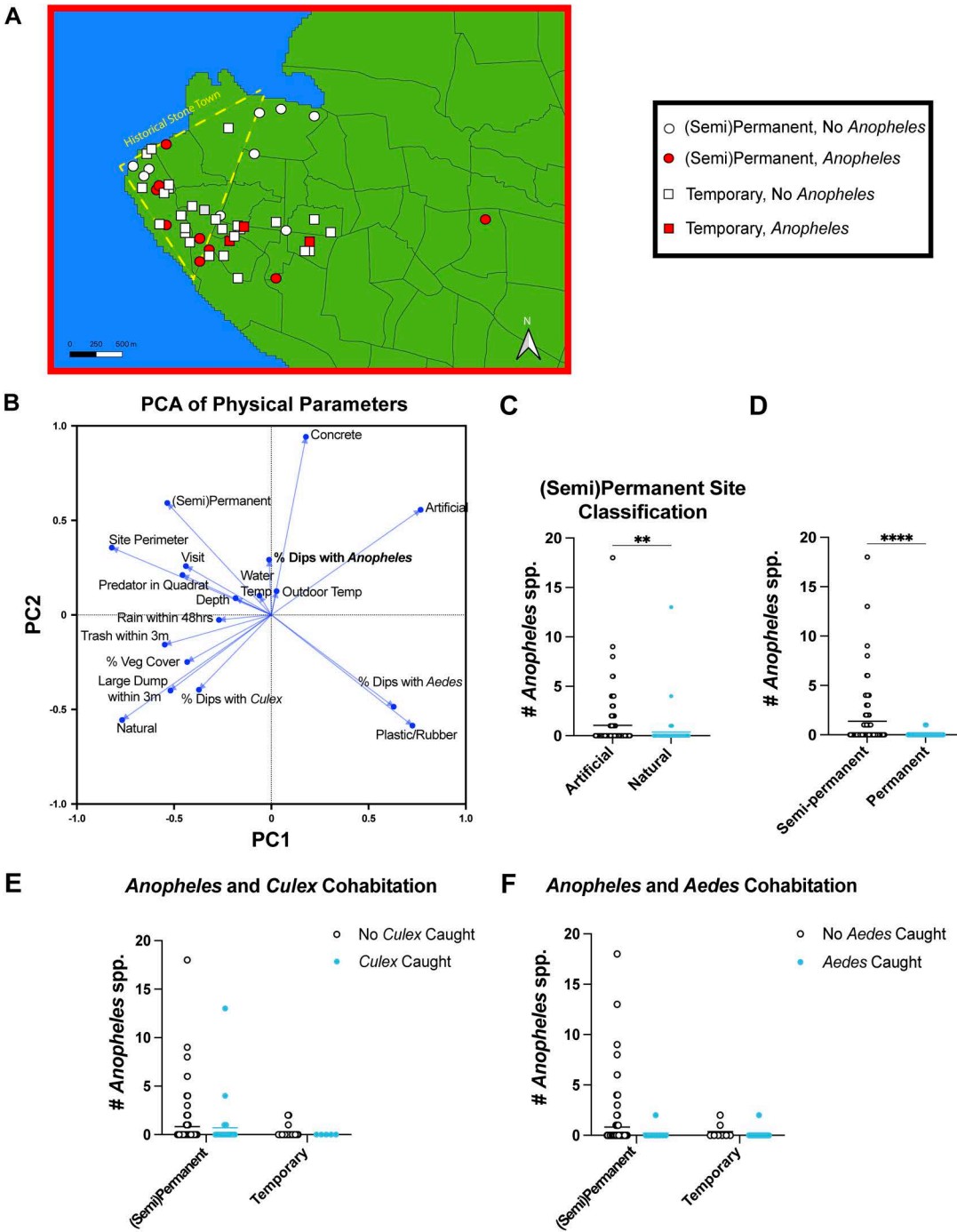

**Fig 2. Characterization of physical qualities of *Anopheles* breeding sites in Zanzibar City, Tanzania. A)** Map showing site permanence and whether site contained *Anopheles* larvae on either visit. Map generated using shapefile provided by GADM under a CC BY license. The shapefile can be found at https://gadm.org/download_country.html. **B)** Principal component analysis graph showing loadings of physical parameters. **C)** Distribution of *Anopheles* larvae abundance in artificial and natural subsites. **D)** Distribution of *Anopheles* larvae abundance in semi-permanent and permanent subsites. **E)** Distribution of *Anopheles* larvae abundance in subsites of different site types in the presence or absence of *Culex* larvae. **F)** Distribution of *Anopheles* larvae abundance in subsites of different site types in the presence or absence of *Aedes* larvae. Statistical significance determined using a Mann-Whitney Test. For all pairwise comparisons: * $P < 0.05$, ** $P < 0.01$, *** $P < 0.001$, **** $P < 0.0001$.

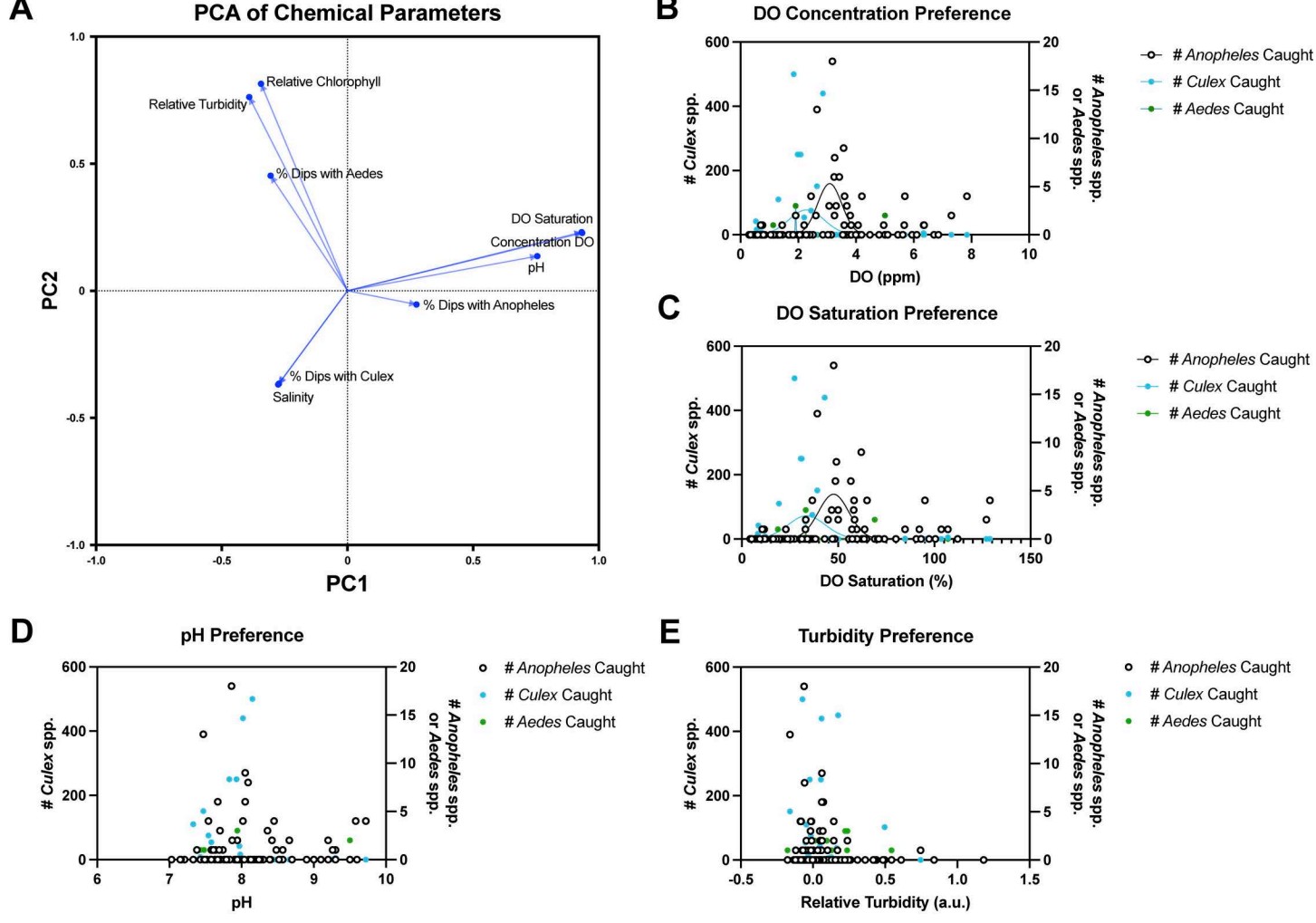

**Fig 3. Characterization of chemical parameters of *Anopheles* breeding sites in Zanzibar City, Tanzania. A**) Principal component analysis showing loadings of chemical parameters. **B**) Distribution of *Culex* abundance (left axis) and *Anopheles*/*Aedes* abundance (right axis) at each subsite plotted against dissolved oxygen concentrations. **C**) Distribution of *Culex* abundance (left axis) and *Anopheles*/*Aedes* abundance (right axis) at each subsite plotted against percentages of dissolved oxygen saturation. **D**) Distribution of *Culex* abundance (left axis) and *Anopheles*/*Aedes* abundance (right axis) at each subsite plotted against pH. **E**) Distribution of *Culex* abundance (left axis) and *Anopheles*/*Aedes* abundance (right axis) at each subsite plotted against relative turbidity.

### Incorporating rainfall data into model to predict *Anopheles* presence

Each permanent/semi-permanent site was visited twice during the sampling period, with at least one week separating each visit. Therefore, differences in *Anopheles* larvae abundance between visits were analyzed. Significantly more *Anopheles* larvae were caught per subsite during the second visit of fountains (Two-tailed Mann-Whitney U test, $P = 0.0169$, $U_{(15,15)} = 67.5$) and wetlands (Two-tailed Mann-Whitney U test, $P = 0.0485$, $U_{(22, 22)} = 187.0$) (Fig 4A). No significant difference in *Anopheles* larvae abundance between visits of artificial ponds and ditches was observed (Fig 4A).

There was a significant difference in rainfall between visits of artificial ponds (Wilcoxon matched-pairs signed rank test, $P = 0.00061$, $W = -108.0$) and fountains (Wilcoxon matched-pairs signed rank test, $P = 0.000061$, $W = -120.0$) (Fig 4B). However, because the visits for each type of permanent/semi-permanent site did not occur on the same day (e.g., not

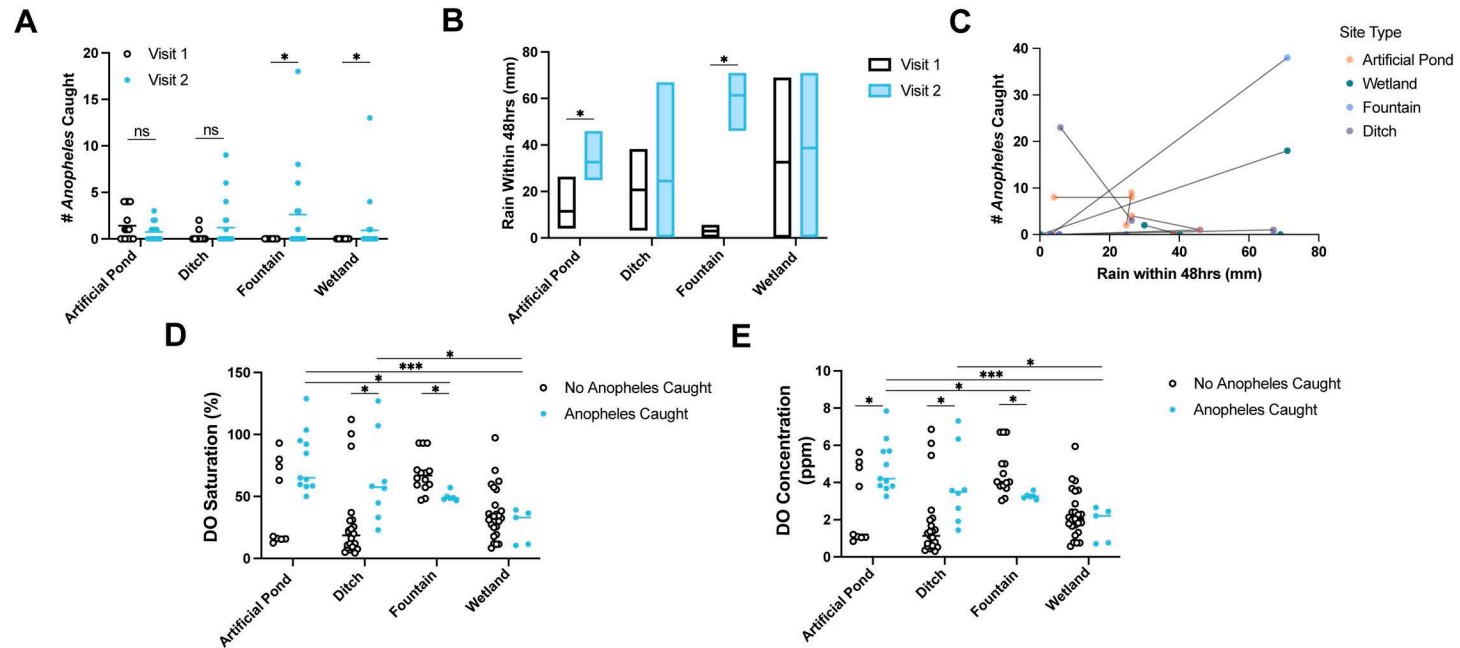

**Fig 4. The effect of rainfall on *Anopheles* breeding site preferences. A)** Distribution of Anopheles larvae abundance at subsites between site types and visits. **B)** Amount of rain within 48hrs of visiting each site, grouped by site types. Statistical significance determined using multiple Wilcoxon tests for paired data. **C)** Individual comparisons of *Anopheles* larvae abundance at each visit to each site. X axis shows the amount of rain within 48hrs of visiting that site. **D)** Distribution of percentage dissolved oxygen saturation across subsites of different site types, split into whether *Anopheles* larvae were caught. **E)** Distribution of dissolved oxygen concentrations across subsites of different site types, split into whether *Anopheles* larvae were caught. Statistical significance was determined using either a Mann-Whitney U Test or Kruskal-Wallis Test with Dunn's uncorrected step-down tests, unless otherwise noted. For all pairwise comparisons: * $P < 0.05$, **$P < 0.01$, ***$P < 0.001$, ****$P < 0.0001$.

all artificial ponds were sampled on the same day), we also compared rainfall between individual sites. Overall, wetlands had significant differences in *Anopheles* abundance between visits but nonsignificant differences in rainfall. However, one wetland had strong, rain-associated increases in *Anopheles* larvae abundance (Fig 4C).

Because PCA indicated an association between *Anopheles* breeding sites in Zanzibar City and higher dissolved oxygen levels, we hypothesized that rain-independent breeding sites might be preferred by *Anopheles* mosquitoes due to consistently higher levels of dissolved oxygen. Additionally, the semi-permanent subsites sampled had significantly greater levels of dissolved oxygen (Welch's T test, $t_{(94.69)} = 6.912$, $P < 0.0001$), lower levels of salinity (Two-tailed Mann-Whitney U test, $U_{(84, 62)} = 2053$, $P = 0.0146$), and fewer predators present than permanent subsites ($x^2 = 5.227$, $df = 1$, $P = 0.0222$) (S2 Fig, S3 and S4 Tables). Commonly observed predators of mosquito larvae included dragonfly/damselfly larvae (order Odonata), backswimmers and water skaters (order Hemiptera), and stonefly larvae (order Plecoptera) (S3 Table). The subsites from artificial ponds and ditches where *Anopheles* larvae were caught had significantly higher levels of dissolved oxygen than the subsites where no *Anopheles* larvae were caught (Fig 4D-E) (S5 and S6 Tables). Subsites from fountains where *Anopheles* larvae were caught had significantly lower levels of dissolved oxygen than subsites from fountains without *Anopheles* larvae, and there was no significant difference in dissolved oxygen levels between either group of subsites from wetlands (S5 and S6 Tables). Subsites with *Anopheles* larvae from artificial ponds had significantly higher levels of dissolved oxygen than subsites with *Anopheles* larvae from fountains and wetlands (Fig 4D-E) (S7 and S8 Tables). Additionally, subsites with *Anopheles* larvae from ditches had significantly higher dissolved oxygen levels than subsites with *Anopheles* larvae from wetlands (S7 and S8 Tables).

## Regression analysis

The data suggested an interaction between dissolved oxygen levels (both saturation and concentration), rainfall within 48hrs, and/or site type, so we fit a logistic regression model to all data from permanent/semi-permanent sites and temporary sites. The outcome used for the logistic regression model was either "*Anopheles* caught" or "no *Anopheles* caught". We included physical parameters that were significant in the original PCA regression analysis, as well as dissolved oxygen concentration and depth. We omitted data that were included in the initial PCA regression analysis, but for which dissolved oxygen data was unavailable. Dissolved oxygen concentration was included in the model instead of dissolved oxygen saturation because dissolved oxygen concentration was more significantly associated with *Anopheles* presence than saturation (as shown in Fig 4D-E). Interaction terms included dissolved oxygen saturation and semi-permanence, dissolved oxygen concentration and concrete, dissolved oxygen concentration and rainfall within 48hrs, concrete and rain within 48hrs, and natural and rain within 48hrs. Significant physical predictors included the percentage of dips with *Aedes* larvae (lower odds of catching *Anopheles* larvae), increased depth (lower odds of catching *Anopheles* larvae), the presence of a predator in the quadrat (lower odds of catching *Anopheles* larvae), and more rain within 48hrs (lower odds of catching *Anopheles* larvae) (Table 1). Significant chemical predictors included increased salinity (lower odds of catching *Anopheles* larvae) (Table 1). Significant interaction terms included increased concentration dissolved oxygen in semi-permanent subsites (lower odds of catching *Anopheles* larvae), increased concentration dissolved oxygen in concrete subsites (higher odds of catching *Anopheles* larvae), and increased concentration of dissolved oxygen after more rainfall within 48hrs (higher odds of catching *Anopheles* larvae) (Table 1). The area under the receiver operating characteristic (ROC) curve was 0.9034, and the positive predictive power was 65.7% (S3 Fig).

## Discussion

In this study, we characterized the breeding sites of *Anopheles* mosquitoes during the rainy season in Zanzibar City. Our results suggest that *Anopheles* breeding sites in Zanzibar City tend to be concrete, semi-permanent water bodies with high dissolved oxygen saturation regardless of heavy rainfall. However, *Anopheles* breeding sites can expand to natural semi-permanent water bodies with high dissolved oxygen concentration after heavier rains.

**Table 1. Logistic regression model of *Anopheles* larvae caught vs. not caught.**

| Parameter estimates | Variable | Odds Ratio | Wald Statistic | P-value |
|---|---|---|---|---|
| β2 | % Dips with Aedes | 0.9222 | 2.406 | 0.0161 |
| β5 | Depth (cm) | 0.9195 | 2.425 | 0.0153 |
| β6 | Salinity | 0.0008745 | 3.674 | 0.0002 |
| β7 | Predator in Quadrat | 0.07624 | 2.349 | 0.0188 |
| β11 | Rain within 48hrs | 0.8722 | 2.408 | 0.016 |
| β12 | (Semi)permanent: Concentration DO | 0.01612 | 2.1 | 0.0357 |
| β13 | Concentration DO: Concrete | 22.91 | 2.328 | 0.0199 |
| β14 | Concentration DO: Rain within 48hrs | 1.076 | 3.376 | 0.0007 |
| Intercept-only AICc | | 145.7 | | |
| Model AICc | | 123.3 | | |
| Area Under ROC Curve | | 0.9034 | | <0.0001 |
| Positive Predictive Power (%) | | 65.71 | | |
| Negative Predictive Power(%) | | 88.76 | | |

Table showing variables with significant odds ratios along with overall model significance and predictive power. Statistical significance determined using a P-value calculated from the t-ratio and number of degrees of freedom. For all statistical comparisons: * $P<0.05$, **$P<0.01$, ***$P<0.001$, ****$P<0.0001$.

Semi-permanent, artificial sites are likely preferred by *Anopheles* mosquitoes over temporary breeding sites due to their longer developmental times compared to larvae from different mosquito genera. *Anopheles* mosquitoes develop from egg to adult in 10–14 days, while *Aedes* and *Culex* mosquitoes develop from eggs to adults in 7–10 days [40]. Since *Anopheles* larvae take longer to develop, they may either be outcompeted in temporary environments where *Aedes* mosquitoes are more suited to breed in, or the container is emptied too frequently for *Anopheles* larvae to develop [30]. This finding is consistent with studies in Ethiopia and elsewhere where more *Anopheles* larvae were found mostly in artificial, permanent/semi-permanent sites [22,41–43]. However, this finding contradicts other studies where *Anopheles* larvae were found mostly in either natural or temporary environments [20,44,45], suggesting either context-dependent adaptations to use semi-permanent sites for breeding, or variation in the definition of "temporary" between studies.

Semi-permanent subsites sampled had more favorable conditions for *Anopheles* than permanent sites, including higher levels of dissolved oxygen, fewer predators, and lower salinity (S2 Fig). Chemical qualities of water bodies influence *Anopheles* ability to breed due to impacts on nutrient availability, toxicity, selection for predators, algae growth, exposure to sunlight, and other factors [46–48]. In Zanzibar City, dissolved oxygen, low abundance of predators, and low salinity are associated with higher *Anopheles* larvae abundance, as seen in some locations but not in others [49,50]. While the reasons behind these differences in breeding site chemical characteristics are likely to be site-specific, permanent sites may have more artificial introduction of predators or competing species, more accumulated pollution than the more frequently drained semipermanent sites, or more nutrients that support a naturally higher population of predators. Higher predator abundance has been associated with fewer *Anopheles* larvae in permanent sites in Ethiopia [48], which is consistent with our data, except semipermanent sites are used as breeding sites in Zanzibar City instead of temporary sites. Therefore, the data presented in this study support a location-dependent hypothesis for *Anopheles* breeding site preferences, where *Anopheles* prefer artificial, semipermanent breeding sites during the rainy season in Zanzibar City.

As high dissolved oxygen is an indicator of overall clean water [51], it is unclear from our data whether *Anopheles* larvae are found in sites with relatively higher dissolved oxygen concentrations/saturation because of the requirement for dissolved oxygen itself, or because of the lack of pollutants that negatively covary with dissolved oxygen levels. Data from experiments on *Culex* and *Aedes* larvae suggest that larvae from these genera can use dissolved oxygen for respiration when reaching the water surface to access atmospheric oxygen is difficult [52,53]. Although formal dissolved oxygen utilization experiments with *Anopheles* larvae are lacking, some authors suggest that dissolved oxygen might itself be especially important for *Anopheles* larvae development because *Anopheles* larvae lack breathing tubes [54]. Alternatively, as higher dissolved oxygen is an indicator of a less-polluted environment [51], the tendency for *Anopheles* breeding sites to have higher dissolved oxygen levels may be solely because those sites are less likely to have harmful pollutants and more likely to have the nutrients required for *Anopheles* development. Due to the lack of a strong, direct correlation between dissolved oxygen levels and *Anopheles* larvae abundance, the data from this study more strongly support the latter hypothesis (especially since the water surface was accessible to larvae in all breeding sites). However, more laboratory-based experimental studies are necessary to determine the role of dissolved oxygen in *Anopheles* larvae development.

Although we observed cohabitation between *Anopheles* and *Culex* larvae, this cohabitation was relatively uncommon, suggesting that each genus occupies distinct niches in Zanzibar City. This lack of common cohabitation is consistent with data from other studies, suggesting that *Culex* and *Anopheles* larvae compete for nutrients [48,55]. The site in our study with the most cohabitation between *Anopheles* and *Culex* larvae was a large, flooded wetland only after heavy rain, which could mean that the rain-associated expansion of this breeding site minimized the competition between mosquito genera. *Culex* larvae are well known to tolerate heavily polluted sites and are often able to avoid competition at these sites [56,57], while the tolerance of *Anopheles* larvae for polluted sites is more debated. While some study locations support the hypothesis that *Anopheles* primarily breed in clean sites [49,50], studies on mainland Africa have provided evidence of *Anopheles* adapting to more polluted environments [41,44,58,59]. With *Aedes* larvae dominating temporary breeding sites and *Culex* larvae dominating more polluted breeding sites with salinity levels above 0 ppt, it is possible that there is

little pressure for *Anopheles* to use more polluted sites for breeding during the rainy season in Zanzibar City. However, as breeding site characteristics change during the dry season - when the semipermanent sites may not be available for breeding [42] - it is likely that *Anopheles* will need to find other breeding sites and may be forced into more polluted environments.

Additionally, it is likely that larval density decreases during the dry period, as other studies have shown higher densities of *Anopheles* mosquitoes during or immediately after rainy seasons [60–62]. However, a recent analysis of *Anopheles stephensi* breeding patterns revealed that seasonal shifts in breeding patterns are also dependent on the degree of urbanization [63], highlighting how seasonal breeding patterns in one location cannot be extrapolated to other locations. Studies of *Anopheles* breeding sites in Zanzibar City during other seasons are, therefore, crucial for a complete understanding of breeding dynamics.

Because this sampling event took place during April (i.e., in rainy season), when consistent rain over many days is followed by days of intense sun, we hypothesized that rain might play a role in the changes in breeding site productivity between visits. Studies have shown that rainfall contributes to either the expansion of *Anopheles* breeding sites, or the flooding of breeding sites and increased mortality of *Anopheles* larvae [64,65]. However, these two outcomes are clearly dependent on the amount of rainfall and the specific shape of the site. The sites in our study most subjected to incoming drainage that could potentially transfer *Anopheles* larvae from other breeding sites were those classified as drains. Additionally, the wetland located at the eastern extreme of the study area also accepted an influx of drainage. While a small number of *Anopheles* larvae were found at the wetland on the eastern extreme of the study area, it is unlikely that the presence of *Anopheles* larvae at this site was dependent on drainage influx because the *Anopheles* were found during the visit with less rain. All other sites were filled mostly by rainwater and did not accept runoff from other locations, so any increase in *Anopheles* larvae can be attributed to breeding at that site. Our data, therefore, suggest that higher rainfall levels are associated with more *Anopheles* larvae being found in wetlands and fountains, while artificial ponds and ditches serve as *Anopheles* breeding sites regardless of rain levels.

Many insects observed alongside mosquito larvae were known predators of mosquito larvae [66–68]. Because predator abundance decreased the odds of observing Anopheles larvae in a subsite, promoting the development of predatory larvae may also help reduce *Anopheles* mosquito abundance. Although their abundance was not recorded in this study, Cyclopoid copepods are also known to be predators of mosquito larvae and have been shown to complement the activity of antilarval phytochemicals [69,70]. Future studies aiming to assess whether copepod abundance correlates with less *Anopheles* breeding site productivity could also help inform antimalarial strategies. Introduction of predatory fish (e.g., *Gambusia* spp.) into artificial ponds has shown to be an effective antilarval strategy in some sites [71]. However, *Gambusia* spp. target other prey over mosquito larvae and are destructive towards the natural ecosystem when introduced [72]. The presence of other prey besides mosquito larvae in a breeding site also affects the practicality of introducing predatory invertebrates into water bodies as an antimalarial strategy [72,73]. Therefore, while introducing natural predators of mosquito larvae may seem like a promising antimalarial strategy, each site's ecosystem must first be analyzed to ensure specific targeting of mosquito larvae. Nevertheless, the introduction of predators could be effective when used in combination with other larvicidal strategies, especially because adult *Anopheles* mosquitoes may be able to detect the presence of predators in water bodies and avoid breeding in such sites (mostly through chemosensory mechanisms) [69]. Future studies are necessary to determine the most effective larvicidal strategy in Zanzibar City.

Although larvicides are rarely used in Africa [23,24], the data from our study suggest that semipermanent ponds or obstructed concrete ditches with high levels of dissolved oxygen saturation may be effective targets for larvicidal strategies in Zanzibar City. Additionally, preventing the flooding and expansion of wetland sites during rain by increasing drainage from those sites might also help prevent *Anopheles* from using these as breeding sites. Because mosquito larvae require stagnant water to breathe [74], larvae abundance in fountains could be minimized by ensuring that these fountains remain functional, and larvae abundance in drains could be minimized by ensuring that these drains remain free of

stagnant water. The use of synthetic larvicides should be avoided, as their use can result in harmful, off-target effects to the environment, and mosquito larvae can develop resistance to the larvicides [75]. However, phytochemicals from plant extracts may be a useful alternative, as they are more benign to the environment, more biodegradable, and contain a diverse array of larvicidal secondary compounds that minimize the risk of resistance [76,77]. Additionally, bacterial larvicides using larva-specific toxins from *Bacillus thuringiensis israelensis* or *Bacillus sphaericus* may be another safe and effective larvicidal strategy [78,79].

Limitations to this study include other potentially important parameters (e.g., algae cover) not being analyzed, limited geographical range, potential differences in *Anopheles* pupae abundance being overlooked, and a lack of species-level resolution of mosquito breeding habits. Differences in microbiota composition between sites were also not analyzed, which could also impact the ability for *Anopheles* larvae to develop [80]. While the purpose of this study was to analyze the *Anopheles* genus, as all *Anopheles* species in Zanzibar are capable of transmitting malaria, individual species have distinct breeding habits in other locations [21,62,81]. Determining which *Anopheles* species are occupying each type of breeding site in Zanzibar City would be useful for monitoring adaptations to more polluted breeding sites and the relative importance of each species in malaria and filarial disease transmission [35,44]. Additionally, larvicidal strategies tend to be more effective in areas where *Anopheles* adults are exophilic, as residual indoor spraying or bed nets are less effective in those areas [82]. Therefore, more recent studies on the behavior of adult *Anopheles* mosquitoes in Zanzibar City are necessary and can also help further assess breeding site productivity [83].

Nevertheless, the short-term nature of the study provides evidence of rapid, rain-associated changes in *Anopheles* breeding site preferences in Zanzibar City, while also revealing sites that are likely to be more consistent breeding sites if they remain filled with water. The small geographical range included provides a targeted analysis of Zanzibar City-specific breeding habitats of *Anopheles* mosquitoes. Additionally, while our study did not include all potentially relevant parameters, the parameters measured are easy to monitor (e.g., more easily than site-specific microbiota) and can therefore be put into practice for larvicidal strategies targeted against *Anopheles*. This study can serve as a baseline for future longitudinal studies on *Anopheles* breeding sites in Zanzibar City, which could be used to develop a larvicidal strategy to combat persistent malaria in the Zanzibar archipelago.

## Conclusion

The incidence of malaria in Zanzibar City has remained between 1–2% since 2015, with a slight but steady increase due to continued import of *Plasmodium* spp. from other malaria endemic areas. Therefore, targeting the resident population of *Anopheles* vector mosquitoes in the largest city of the Zanzibar archipelago is a crucial strategy to combating malaria on the islands. With pyrethroid resistance in adult *Anopheles* mosquitoes increasing on Unguja, larvicidal strategies could serve as an alternative strategy to decrease the population numbers of resident *Anopheles* mosquitoes. Although the geographic range of this study focused on the core of Zanzibar City, *Anopheles* larvae appeared to be evenly distributed between core Stone Town and the surrounding Zanzibar City. However, as most beautification projects and artificial, semipermanent structures are in the more tourist-trafficked area of core Stone Town, careful monitoring of these projects is necessary to prevent these concrete basins in parks from becoming nutrient and oxygen-rich breeding grounds for *Anopheles* mosquitoes.

Our study provides the first systematic survey of *Anopheles* breeding sites in Zanzibar City. Our findings that *Anopheles* prefer either semipermanent, concrete habitats with high dissolved oxygen levels, but can expand to sites with lower dissolved oxygen levels after heavy rains, suggest that *Anopheles* larvae may be targeted as an antimalarial strategy. However, because our study is the first of its kind conducted in Zanzibar City, future longitudinal research is required to characterize the breeding sites of *Anopheles* mosquitoes during dry periods. Additionally, the productivity of each breeding site should be studied by analyzing which breeding sites are producing *Anopheles* pupae and mature mosquitoes. Given the current targetability of *Anopheles* larvae, potential public health policies that may reduce the population of *Anopheles*

mosquitoes during the rainy season could include introducing phytochemical larvicides into artificial ponds, ensuring that fountains are functional, and preventing the flooding of wetlands. Through a combination of strategies targeting larval and adult *Anopheles*, the resident population of *Anopheles* mosquitoes could be depleted sufficiently to eliminate malaria in Unguja.

## Supporting information

**S1 Fig. Photos of examples of each site type. A**) Artificial pond. **B**) Ditch. **C**) Fountain. **D**) Wetland. **E**) Temporary water jug. **F**) Temporary tire.
(TIF)

**S2 Fig. Differences in water quality parameters between semi-permanent and permanent subsites. A**) Dissolved oxygen concentrations in semi-permanent and permanent subsites. **B**) Dissolved oxygen saturation percentages in semi-permanent and permanent subsites. **C**) Salinity levels in semi-permanent and permanent subsites. Statistical significance determined using Student's unpaired T tests or Mann-Whitney tests for nonparametric data (salinity levels). For all pairwise comparisons: * $P < 0.05$, **$P < 0.01$, ***$P < 0.001$, ****$P < 0.0001$.
(TIF)

**S3 Fig. Receiver operator characteristic curve for the logistic regression model.** Curve shows a deviation of the current model (black line) from a model with 50/50 odds of predicting *Anopheles* presence (red line).
(TIF)

**S1 Table. Permanent/semi-permanent site-level *Anopheles* larvae abundances.**
(PDF)

**S2 Table. Principal component regression results for physical parameters.** All slope estimates are presented in terms of the parameters, not in terms of principal components. Statistical significance for each parameter determined automatically using a T-distribution. Overall significance for the regression determined using ANOVA. For all statistical comparisons: * $P < 0.05$, **$P < 0.01$, ***$P < 0.001$, ****$P < 0.0001$.
(PDF)

**S3 Table. Predatory taxa found at each site.**
(PDF)

**S4 Table. Differences in predator abundance between semi-permanent and permanent subsites.** Statistical significance determined using Chi-square test for independence.
(PDF)

**S5 Table. Two-tailed Mann-Whitney U test results comparing oxygen concentration between subsites with and without *Anopheles* larvae, split by site type.** Only significant pairwise comparisons shown.
(PDF)

**S6 Table. Two-tailed Mann-Whitney U test results comparing oxygen saturation between subsites with and without *Anopheles* larvae, split by site type.** Only significant pairwise comparisons shown.
(PDF)

**S7 Table. Kruskal-Wallace test results comparing dissolved oxygen concentration at subsites with *Anopheles* between different site types.** Only significant pairwise comparisons shown.
(PDF)

**S8 Table. Kruskal-Wallace test results comparing dissolved oxygen saturation levels at different site types.** Only significant pairwise comparisons shown.
(PDF)

**S1 File. Inclusivity in global research questionnaire.**
(DOCX)

**S2 File. Raw data.**
(CSV)

## Acknowledgments

We would like to acknowledge and thank the faculty and staff at the School for International Training office in Zanzibar for providing logistical and material help for this project. We would like to thank Dr. Mohamed Jiddawi at the Zanzibar Medical Group for the thoughtful advice when conceptualizing the study. Additionally, we thank Dr. Mohammed Maalim at the University of Dar es Salaam Institute of Marine Sciences for lending us the water quality measurement devices.

## Author contributions

**Conceptualization:** Kaeden Hill, Narriman S. Jiddawi, Jonathan R. Walz, Katharina Kreppel.

**Data curation:** Kaeden Hill, Dickson Kobe.

**Formal analysis:** Kaeden Hill.

**Investigation:** Kaeden Hill, Dickson Kobe.

**Methodology:** Kaeden Hill, Dickson Kobe, Narriman S. Jiddawi, Jonathan R. Walz, Katharina Kreppel.

**Project administration:** Narriman S. Jiddawi, Jonathan R. Walz, Katharina Kreppel.

**Resources:** Kaeden Hill, Jonathan R. Walz.

**Supervision:** Dickson Kobe, Narriman S. Jiddawi, Jonathan R. Walz, Katharina Kreppel.

**Validation:** Kaeden Hill.

**Visualization:** Kaeden Hill.

**Writing – original draft:** Kaeden Hill.

**Writing – review & editing:** Kaeden Hill, Narriman S. Jiddawi, Jonathan R. Walz, Katharina Kreppel.

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
