## [Decision Letter · Decision Letter 0]

10 Jan 2025

PONE-D-24-47727Larval surveys reveal breeding site preferences of malaria vector Anopheles spp. in Zanzibar CityPLOS ONE

Dear Dr. Hill,

Thank you for submitting your manuscript to PLOS ONE. After careful consideration, we feel that it has merit but does not fully meet PLOS ONE’s publication criteria as it currently stands. Therefore, we invite you to submit a revised version of the manuscript that addresses the points raised during the review process.

We look forward to receiving your revised manuscript.

Kind regards,

Jiang-Shiou Hwang, Ph.D.

Academic Editor

PLOS ONE

Journal Requirements:

2. Please include a complete copy of PLOS’ questionnaire on inclusivity in global research in your revised manuscript. Our policy for research in this area aims to improve transparency in the reporting of research performed outside of researchers’ own country or community. The policy applies to researchers who have travelled to a different country to conduct research, research with Indigenous populations or their lands, and research on cultural artefacts. The questionnaire can also be requested at the journal’s discretion for any other submissions, even if these conditions are not met.  

Please find more information on the policy and a link to download a blank copy of the questionnaire here: https://journals.plos.org/plosone/s/best-practices-in-research-reporting. 

Please upload a completed version of your questionnaire as Supporting Information when you resubmit your manuscript.

3. Thank you for stating the following in your manuscript: 

“FUNDING

This project was funded by personal funds from KH and School for International

Training study abroad program funds.”

4. We note that Figures 1 and 2 in your submission contain map/satellite images which may be copyrighted. All PLOS content is published under the Creative Commons Attribution License (CC BY 4.0), which means that the manuscript, images, and Supporting Information files will be freely available online, and any third party is permitted to access, download, copy, distribute, and use these materials in any way, even commercially, with proper attribution. For these reasons, we cannot publish previously copyrighted maps or satellite images created using proprietary data, such as Google software (Google Maps, Street View, and Earth). For more information, see our copyright guidelines: http://journals.plos.org/plosone/s/licenses-and-copyright.

1) You may seek permission from the original copyright holder of Figures 1 and 2 to publish the content specifically under the CC BY 4.0 license.  

2) If you are unable to obtain permission from the original copyright holder to publish these figures under the CC BY 4.0 license or if the copyright holder’s requirements are incompatible with the CC BY 4.0 license, please either i) remove the figure or ii) supply a replacement figure that complies with the CC BY 4.0 license. Please check copyright information on all replacement figures and update the figure caption with source information. If applicable, please specify in the figure caption text when a figure is similar but not identical to the original image and is therefore for illustrative purposes only.

5. We notice that your supplementary figures are uploaded with the file type 'Figure'. Please amend the file type to 'Supporting Information'. Please ensure that each Supporting Information file has a legend listed in the manuscript after the references list.

6. We notice that your supplementary tables are included in the manuscript file. Please remove them and upload them with the file type 'Supporting Information'. Please ensure that each Supporting Information file has a legend listed in the manuscript after the references list. 

Reviewers' comments:

Reviewer's Responses to Questions

**Comments to the Author**

1. Is the manuscript technically sound, and do the data support the conclusions?

Reviewer #1: Yes

Reviewer #2: Yes

2. Has the statistical analysis been performed appropriately and rigorously? 

Reviewer #1: Yes

Reviewer #2: Yes

3. Have the authors made all data underlying the findings in their manuscript fully available?

Reviewer #1: Yes

Reviewer #2: Yes

4. Is the manuscript presented in an intelligible fashion and written in standard English?

Reviewer #1: Yes

Reviewer #2: Yes

5. Review Comments to the Author

Reviewer #1: The article is well structured, introducing the global burden of malaria, contextualizing it in Zanzibar City, and detailing the study's methods and findings. The study addresses an important public health issue and provides actionable insights for malaria control in Zanzibar City. The use of statistics strengthens the argument and provides concrete evidences.

However, reference formatting is needed as per journal requirement. Article needs to address grammatical and typographical errors thoroughly. At many places the meaning of the sentences is not clear. Such sentences can be reframed or break into two separate sentences for better readability. The conclusion could be strengthened by briefly outlining the practical implications of the findings for public health policy and future research.

In the discussion, 3-4 sentences on biocontrol agents for mosquito larvae can be included while addressing the study's limitations. Few relevant references can be cited in the study

How effective are Mesocyclops aspericornis (Copepoda: Cyclopoida) in controlling mosquito immatures in the environment with an application of phytochemicals?

R Dhanker, R Kumar, JS Hwang

Hydrobiologia 716, 147-162

Larvicidal efficiency of aquatic predators: a perspective for mosquito biocontrol - JSH Ram Kumar, Zoological Studies 45 (4), 447

Manuscript can be accepted after Minor Revision.

Reviewer #2: This is the firt study from this region, which needs special attention because of unsuccessful malaria control programme. The incidence of malaria 1-2%, with an increasing trend lately demands multiple control strategies including proper identification of breeding sites of Anopheles. Thus this study is useful and conducted with proper objective and well designed protocol for site selection . Authors should explicitly deliberated which are requires special care and was any specific larvivorous organisms that showed significant correlation with larval abundance. My specific comments are as follow:

Many sentences are verbose need restructuring

in abstract

Line 30-44: unnecessary background information not required in abstract. These dilute the essence of present results.

Line 33: vector is not used as verb better to say transmitted

Line 36-39: Restructure

Introduction:

Line 57-59 Too broad and too general statement not required in such data riven paper.

Line 63: Understanding the ecology -" shift after line 74.

Line 79-80: replace which type of breeding sites with attributed of breeding sites and hydro-period

Line 83 to 118 These explanation should be provided in discussion and some parts in Introduction before line 79 -i.e. before stating the goal of this paper as Therefore, the aim of this study was to determine which type of breeding sites are used by Anopheles mosquitoes during the rainy season in Zanzibar City and what characterizes them by identifying the physicochemical parameters.

Discussion need reorientation focusing major results, inferences and concurring predators i.e. larvicidal organisms in these habitats, see following papers for larvicidal organisms

1. DOI: 10.1186/s40555-015-0132-9

2. How effective are Mesocyclops aspericornis (Copepoda: Cyclopoida) in controlling mosquito immatures in the environment with an application of phytochemicals? October 2013 , Hydrobiologia

3. Potential of three aquatic predators to control mosquitoes in the presence of alternative prey: A comparative experimental assessment, January 2008; Marine and Freshwater Research 59(9). DOI: 10.1071/MF07143

4. Larvicidal efficiency of aquatic predators: A perspective for mosquito biocontrol. 2006 Zoological Studies

5. Predation on Mosquito Larvae by Mesocyclops thermocyclopoides (Copepoda: Cyclopoida) in the Presence of Alternate Prey 2003 Internationale Revue der gesamten Hydrobiologie und Hydrographie 88(6):570 - 581

DOI: 10.1002/iroh.200310631

Further discussion should flow around hydro-period, and larval-pupal period in water and breeding by adult Anopheles,

Water quality parameters.

Table 1 is very lengthy only significant values may be shown.

Line 610-612: environmentally friendly is very broad and vague statement so explicitly mention what control strategy is recommended by this study , operational control programm, ecology of breeding habitat and recurrence of malaria and

6. PLOS authors have the option to publish the peer review history of their article (what does this mean? ). If published, this will include your full peer review and any attached files.

**Do you want your identity to be public for this peer review?** For information about this choice, including consent withdrawal, please see our Privacy Policy .

Reviewer #1: No

Reviewer #2: No

---

## [Author Response · Author response to Decision Letter 0]

25 Feb 2025

Reviewer 1:

1. “Article needs to address grammatical and typographical errors thoroughly. At many places the meaning of the sentences is not clear. Such sentences can be reframed or break into two separate sentences for better readability.”

Response: We have shortened sentences throughout the manuscript to promote better readability.

2. “The conclusion could be strengthened by briefly outlining the practical implications of the findings for public health policy and future research.”

Response: We thank the reviewer for this suggestion. We have updated the conclusion to address the value of future studies aiming to characterize the seasonal variability of Anopheles breeding preferences and determine the productivity of each breeding site. Additionally, we restructured the conclusion to end with a brief list of potential policies (supported by our data) that might help reduce Anopheles breeding in Zanzibar City.

3. “In the discussion, 3-4 sentences on biocontrol agents for mosquito larvae can be included while addressing the study's limitations. Few relevant references can be cited in the study:

a. How effective are Mesocyclops aspericornis (Copepoda: Cyclopoida) in controlling mosquito immatures in the environment with an application of phytochemicals? R Dhanker, R Kumar, JS Hwang, Hydrobiologia 716, 147-162

b. Larvicidal efficiency of aquatic predators: a perspective for mosquito biocontrol - JSH Ram Kumar, Zoological Studies 45 (4), 447”

Response: We thank the reviewer for this suggestion and for these sources. We have incorporated these sources and added a more in-depth discussion of how predators could be used alongside other strategies to control mosquito larvae numbers. As we did not assess copepod abundance in our study, we stress that copepods should be identified alongside other predators during future larval surveys. To better inform future larval surveys or larvicidal strategies in Zanzibar City, we have provided a more detailed depiction of the types of mosquito larvae predators observed in our study (shown below and added to the manuscript as S3 Table).

Reviewer 2:

1. “Many sentences are verbose need restructuring”

Response: We have shortened sentences throughout the article (please see tracked changes document).

2. “Line 30-44: unnecessary background information not required in abstract. These dilute the essence of present results.”

Response: We thank the reviewer for this suggestion. We have shortened the abstract by removing the first three sentences that introduced the reader to malaria, as this information is obtained in the introduction. The updated abstract begins by telling the reader why larval surveys are needed given the rising trend in malaria cases in Zanzibar City.

3. “Line 33: vector is not used as verb better to say transmitted”

Response: This sentence was removed from the abstract.

4. “Line 36-39: Restructure”

Response: We have restructured these lines of the abstract to minimize complex sentence structure.

5. “Line 57-59 Too broad and too general statement not required in such data riven paper.”

Response: We have restructured this sentence to only include background information on how malaria is transmitted. We want to ensure that this information is presented to readers, as it is crucial to understand the argument of our paper.

6. “Line 63: Understanding the ecology -" shift after line 74.”

Response: We thank the reviewer for this suggestion. The sentence on line 63 is now at line 68, after we introduce the need for alternative antimalarial strategies.

7. “Line 79-80: replace which type of breeding sites with attributed of breeding sites and hydro-period”

Response: We have restructured this line.

8. “Line 83 to 118 These explanation should be provided in discussion and some parts in Introduction before line 79 -i.e. before stating the goal of this paper as Therefore, the aim of this study was to determine which type of breeding sites are used by Anopheles mosquitoes during the rainy season in Zanzibar City and what characterizes them by identifying the physicochemical parameters.”

Response: The introduction has been restructured to address this comment. We now discuss the aims of our study in the last paragraph of the introduction.

9. “Discussion need reorientation focusing major results, inferences and concurring predators i.e. larvicidal organisms in these habitats, see following papers for larvicidal organisms

a. DOI: 10.1186/s40555-015-0132-9

b. How effective are Mesocyclops aspericornis (Copepoda: Cyclopoida) in controlling mosquito immatures in the environment with an application of phytochemicals? October 2013 , Hydrobiologia

c. Potential of three aquatic predators to control mosquitoes in the presence of alternative prey: A comparative experimental assessment, January 2008; Marine and Freshwater Research 59(9). DOI: 10.1071/MF07143

d. Larvicidal efficiency of aquatic predators: A perspective for mosquito biocontrol. 2006 Zoological Studies

e. Predation on Mosquito Larvae by Mesocyclops thermocyclopoides (Copepoda: Cyclopoida) in the Presence of Alternate Prey 2003 Internationale Revue der gesamten Hydrobiologie und Hydrographie 88(6):570 – 581

f. DOI: 10.1002/iroh.200310631”

Response: We thank the reviewer for this comment and for providing these sources. We have incorporated these sources into the end of our discussion. Additionally, as mentioned in our response to Comment 3 from Reviewer 1, we have provided more detailed data on the types of predators identified at each site. This table has been added to the manuscript as S3 Table, and the results section has been updated accordingly.

10. “Further discussion should flow around hydro-period, and larval-pupal period in water and breeding by adult Anopheles.”

Response: We thank the reviewer for this suggestion. In addition to our preexisting paragraph on how rain could be either harmful or beneficial to Anopheles breeding (lines 522-539), we have added a discussion on the role that seasonal weather patterns (i.e. hydro-period) might have in Anopheles breeding habits (lines 514-521). This addition to our discussion also addresses your suggestion to further comment on adult Anopheles breeding patterns. We cite a recent study showing that the degree of urbanization might affect the extent that hydro-period influences Anopheles breeding patterns (3). The findings presented in that study may be applicable to Zanzibar City, given its rapidly increasing degree of urbanization. As we did not analyze the abundance of Anopheles pupae in our study, we have included this shortcoming in our discussion of the study’s limitations (line 757-760). Additionally, we added a note on the importance of conducting future studies on Anopheles pupae and adult abundance in order to determine which breeding sites are most effective. We also did not analyze the habits of adult Anopheles mosquitoes in our study, and we addressed this limitation in our discussion (lines 588-592) and conclusion (lines 626-628).

Also, as an additional avenue of further discussing breeding by adult Anopheles, we have added the possibility that Anopheles adults may be able to avoid breeding in sites with high predator abundance through chemical and visual cues (lines 556-559) (4). Therefore, it is crucial for future studies to analyze the dynamics and behavior of adult Anopheles mosquitoes in Zanzibar City to design the most effective larvicidal strategy.

11. “Table 1 is very lengthy only significant values may be shown.”

Response: We have removed non-significant parameters from the table for conciseness.

12. “Line 610-612: environmentally friendly is very broad and vague statement so explicitly mention what control strategy is recommended by this study , operational control programm, ecology of breeding habitat and recurrence of malaria and”

Response: We have removed the term “environmentally friendly.” Instead, we recommend “introducing phytochemical larvicides into artificial ponds, ensuring that fountains are functional, and preventing the flooding of wetlands (lines 629-632).

References

1. Tanzania Districts Shapefiles 2019. (2012).

2. 2022 Population and Housing Census - Tanzania Wards. (2024).

3. C. Whittaker, et al., Seasonal dynamics of Anopheles stephensi and its implications for mosquito detection and emergent malaria control in the Horn of Africa. Proceedings of the National Academy of Sciences 120, e2216142120 (2023).

4. R. Kumar, J.-S. Hwang, Larvicidal efficiency of aquatic predators: A perspective for mosquito biocontrol. Zoological Studies 45, 447–466 (2006).

---

## [Decision Letter · Decision Letter 1]

8 Apr 2025

PONE-D-24-47727R1Larval surveys reveal breeding site preferences of malaria vector *Anopheles* spp. in Zanzibar CityPLOS ONE

Dear Dr. Hill,

Thank you for submitting your manuscript to PLOS ONE. After careful consideration, we feel that it has merit but does not fully meet PLOS ONE’s publication criteria as it currently stands. Therefore, we invite you to submit a revised version of the manuscript that addresses the points raised during the review process.

We look forward to receiving your revised manuscript.

Kind regards,

Jiang-Shiou Hwang, Ph.D.

Academic Editor

PLOS ONE

Journal Requirements:

Reviewers' comments:

Reviewer's Responses to Questions

**Comments to the Author**

1. If the authors have adequately addressed your comments raised in a previous round of review and you feel that this manuscript is now acceptable for publication, you may indicate that here to bypass the “Comments to the Author” section, enter your conflict of interest statement in the “Confidential to Editor” section, and submit your "Accept" recommendation.

Reviewer #1: All comments have been addressed

Reviewer #2: All comments have been addressed

2. Is the manuscript technically sound, and do the data support the conclusions?

Reviewer #1: Yes

Reviewer #2: Yes

3. Has the statistical analysis been performed appropriately and rigorously? 

Reviewer #1: Yes

Reviewer #2: Yes

4. Have the authors made all data underlying the findings in their manuscript fully available?

Reviewer #1: Yes

Reviewer #2: Yes

5. Is the manuscript presented in an intelligible fashion and written in standard English?

Reviewer #1: Yes

Reviewer #2: Yes

6. Review Comments to the Author

Reviewer #1: The authors have thoroughly addressed all the queries. The manuscript can be accepted after removing the citations from the conclusion, as citations are generally not included in this section.

Reviewer #2: The manuscript has been revised very carefully. The revised version is sign substantially improved in clarity, coherence and tone. I appreciate the author’s commitment to produce a high quality manuscript.

I have following minor observation

Line 220 221 Data analysis and statistics: better change to statistical analyses

Line 222: delete was as “All statistical analyses were done using GraphPad Prism 10 (10.1.1).

Table -1 Delete the last column i.e. P-value summary; p values are given very clearly no need to add additional column for summary

Line 522: In correct: Because this sampling period took place during the April rainy season

Correct: Because this sampling event took place during the April i.e. in rainy season

Line 528-537 : Rephrase sentence and split in two sentences: Of all the sites sampled in this study, those most subjected to drainage coming into the site (and potentially bringing Anopheles larvae from other breeding sites) were those classified as drains and the wetland located at the eastern extreme of the study area.

7. PLOS authors have the option to publish the peer review history of their article (what does this mean? ). If published, this will include your full peer review and any attached files.

**Do you want your identity to be public for this peer review?** For information about this choice, including consent withdrawal, please see our Privacy Policy .

Reviewer #1: No

Reviewer #2: No

---

## [Author Response · Author response to Decision Letter 1]

9 Apr 2025

Reviewer #1:

The authors have thoroughly addressed all the queries. The manuscript can be accepted after removing the citations from the conclusion, as citations are generally not included in this section.

Response: We thank the reviewer for this comment. We have removed citations from the conclusion. No new material was referenced in the conclusion.

Reviewer #2:

Line 220 221 Data analysis and statistics: better change to statistical analyses

Response: We thank the reviewer for this comment. We have updated the title of this subsection.

Line 222: delete was as “All statistical analyses were done using GraphPad Prism 10 (10.1.1).

Response: We thank the reviewer for catching this error in our phrasing. We have updated the sentence.

Table -1 Delete the last column i.e. P-value summary; p values are given very clearly no need to add additional column for summary

Response: We have removed this column from Table 1 in the manuscript.

Line 522: In correct: Because this sampling period took place during the April rainy season. Correct: Because this sampling event took place during the April i.e. in rainy season

Response: We thank the reviewer for this suggestion. We have updated this sentence.

Line 528-537 : Rephrase sentence and split in two sentences: Of all the sites sampled in this study, those most subjected to drainage coming into the site (and potentially bringing Anopheles larvae from other breeding sites) were those classified as drains and the wetland located at the eastern extreme of the study area.

Response: We thank the reviewer for this suggestion. We have updated these sentences.

---

## [Editor Report · Decision Letter 2]

11 Apr 2025

Larval surveys reveal breeding site preferences of malaria vector *Anopheles* spp. in Zanzibar City

PONE-D-24-47727R2

Dear Dr. Hill,

We’re pleased to inform you that your manuscript has been judged scientifically suitable for publication and will be formally accepted for publication once it meets all outstanding technical requirements.

Kind regards,

Jiang-Shiou Hwang, Ph.D.

Academic Editor

PLOS ONE
---

## [Editor Report · Acceptance letter]

PONE-D-24-47727R2

PLOS ONE

Dear Dr. Hill,

I'm pleased to inform you that your manuscript has been deemed suitable for publication in PLOS ONE. Congratulations! Your manuscript is now being handed over to our production team.

Kind regards,

on behalf of

Prof. Jiang-Shiou Hwang

Academic Editor

PLOS ONE